# Multiscale Study of Interactions Between Corrosion Products Layer Formed on Heritage Cu Objects and Organic Protection Treatments

**Maëva L'héronde [1,2], Muriel Bouttemy [2], Florence Mercier-Bion [1,*], Delphine Neff [1], Emilande Apchain [1], Arnaud Etcheberry [2] and Philippe Dillmann [1]**

[1]  LAPA-IRAMAT, NIMBE, CEA, CNRS, Université Paris-Saclay, CEA Saclay, 91191 Gif-sur-Yvette, France; maeva.lheronde@synchrotron-soleil.fr (M.L.); delphine.neff@cea.fr (D.N.); emilande.apchain@u-cergy.fr (E.A.); philippe.dillmann@cea.fr (P.D.)

[2]  Institut Lavoisier de Versailles (ILV), Université de Versailles Saint-Quentin-en-Yvelines, Université Paris-Saclay, CNRS, 45 avenue des États-Unis, 78035 Versailles, France; muriel.bouttemy@uvsq.fr (M.B.); arnaud.etcheberry@uvsq.fr (A.E.)

*   Correspondence: florence.mercier@cea.fr

**Abstract:** In the framework of the protection of copper objects exposed to atmospheric corrosion, different solutions are envisaged, among them carboxylate treatments ($HC_{10}$). In this study, an analytical approach based on complementary techniques from micrometer to nanometer scale (μRS, SEM-EDS, SAM) is used to describe the properties of the corrosion products layer (CPL) and determine the penetration depth of the $HC_{10}$ protection treatment inside the CPL of copper samples issued from the roof of the Saint Martin church in Metz. The CPL consists in a thick brochantite layer (20 to 50 μm), mainly composed of $Cu_4SO_4(OH)_6$, on top of a thinner (1 to 5 μm thick) cuprite layer, $Cu_2O$, acting as a natural corrosion barrier on the metal. Application of the organic treatment is implemented by immersing the corroded samples in $HC_{10}$ solution, consistent with future requirements for large scale applications. Even for short-term duration (one minute), the $HC_{10}$ treatment penetrates to the cuprite/brochantite interface, but $Cu(C_{10})_2$ precipitate is only detected locally, whereas for a longer immersion of thirty minutes, it is present in higher proportions in the whole brochantite layer, filling the pores, up to the cuprite/brochantite interface. $Cu(C_{10})_2$ acts as a second inner barrier and prevents liquid infiltration.

**Keywords:** corrosion; copper; cultural heritage; corrosion inhibitor; carboxylate treatment; Auger spectroscopy

## 1. Introduction

In the outdoor environment, copper cultural heritage objects undergo alterations caused by water and atmospheric pollution. The atmospheric corrosion of these objects leads to their physical and aesthetic modifications by the interaction of the environment with the patina. This superficial corrosion layer is finally an integral part of the object and has to be preserved. To protect the cultural heritage objects from the atmospheric corrosion, different types of treatments are used by the restorers: varnishes, waxes, and corrosion inhibitors [1]. The advantage of varnishes and waxes is that they do not significantly modify the surface appearance but are efficient. Indeed, such treatments create a physical barrier on the surface and thus prevent interactions between the atmosphere and the objects. However, varnishes have the disadvantage of being difficult to reprocess and waxes of having a bad hold in time. Concerning corrosion inhibitors, including benzotriazole (BTA) which is currently

used for restoration, several studies have reported a possible toxicity of this type of inhibitor for the environment and, as well, for the restorers [2–4].

The search for new corrosion inhibiting treatments, non-toxic, is a major issue in the field of heritage. So, other types of corrosion inhibitors have emerged since several decades: Carboxylates, derivatives of fatty acids extracted from vegetable oils [5–13]. We have chosen to develop a protection treatment procedure using a carboxylate organic compound. The chemical formula of the carboxylic acids is $CH_3$-$(CH_2)_{n-2}$-COOH, generally denoted by $HC_n$, where n corresponds to the number of carbon atoms composing the aliphatic chain, n being equal to 10 in the present study ($HC_{10}$). The inhibitory property of carboxylate solutions results from the reaction between the copper ions and the carboxylates which leads to the precipitation of metal carboxylates $Cu(C_{10})_2$ (metal soaps) on the surface of the corrosion products layer (CPL). These metallic soaps act as surfactants due to the carboxylate part of the molecule providing a hydrophilic character, but have also hydrophobic properties, in relation with the aliphatic chain. Due to its hydrophobic character, this molecule may form a layer on the surface of the metal, and in this case avoid the penetration of water to the interface with the metal and slows down the corrosion of the copper object. To our knowledge, carboxylate treatments have been mainly studied on non-patinated copper objects. Such treatments have yet proven their efficiency in industry and non-corroded heritage objects (iron and copper) with the evidence of corrosion limitation for different metals (zinc, lead and iron) [5–12].

In recent studies [13], we have succeeded in implementing this type of treatment on corroded samples issued from heritage objects and have highlighted its effectiveness. Present work focuses on the determination of the optimal conditions of use of the carboxylate treatment ($HC_{10}$) and particularly, the immersion time as a key factor. The fine determination of the depth penetration of the $HC_{10}$ solution in the CPL and its interactions with the corrosion products are major issues. Indeed, the literature concerning the treatment of corroded metals by $HC_{10}$ is not extensive, and especially, the CPL/$HC_{10}$ layer interaction mechanism remains an open question. To bring new insights on this problematic, we have developed an analytical approach based on the complementary use of different techniques, with spatial resolution ranging from micrometer to nanometer scale: μRaman Spectroscopy (μRS), Scanning Electron Microscopy (SEM) with Energy Dispersive Spectroscopy (EDS) and Scanning Auger Microscopy (SAM). Such analytical strategy, combining more usual (OM, SEM-EDS, μRS, XRD … ) and advanced characterization techniques (XPS, SIMS, PIXE, RBS … ) using either conventional laboratory or particle accelerator sources has already been employed on Cu based and metallic artefacts to study the corrosion phenomena in various conditions such as sea water immersion, soils burying or air ageing [14–16] enabling to bring an accurate description of the corrosion layers and crucial information about the corrosion mechanism. This knowledge represents an undeniable added value to guide preservation and storage actions. This multi-technique approach is also a good practice to ensure the reliability of the chemical information obtained. Different studies have shown for example the instability of copper compounds under X-Ray irradiation [17] and so evidenced the difficulty to implement XPS without modifying the initial information about the oxidation degree. This specific question will be treated in more details in a future paper.

In the present study, analyses were realized on copper samples issued from the roof of the Saint Martin church in Metz, immersed during one and thirty min in the $HC_{10}$ solution. We have decided to employ chemical analyses only for the determination of the spatial distribution of elements considering tracking elements representative of the different compounds present Cu, O, S for the CPL and C for the $Cu(C_{10})_2$. In particular, SAM implementation, essential in our approach to reach the sub-micron scale, was a real challenge on treated samples containing organic compounds not really prone to be analyzed by a focused electron beam technique and in UHV. However, thanks to a tailored sample preparation protocol, significant information concerning the $Cu(C_{10})_2$ formation could be evidenced bringing a better insight on its action against corrosion. Thanks to this multi-technique approach, CPL composition and structuration could be accurately determined, $HC_{10}$ penetration studied and its

inhibiting capabilities evaluated from 1 to 30 min. This represents an important knowledge for the development and optimization of future preservation strategies.

## 2. Materials and Methods

### 2.1. Samples

The samples were supplied by LRMH Laboratory (Laboratoire de Recherche des Monuments Historiques) and prepared in nearly $1 \times 1$ cm$^2$ square pieces from the copper roof of Saint Martin church in Metz dated of 1877 that suffered long-term atmospheric corrosion, They consist of pure copper containing micrometric inclusions mostly composed of lead, arsenic, but also antimony, nickel and bismuth from SEM-EDS analyses. The side exposed to atmosphere was easily distinguishable by the characteristic pale green corrosion layer present on top. They were cleaned with ethanol and dried at ambient conditions to eliminate the surface contamination of the sample before proceeding to the organic treatment.

### 2.2. Carboxylate Protective Treatment

The protective treatment was realized by immersion in a HC$_{10}$ solution in the laboratory. The solution was prepared by dissolving the decanoic acid HC$_{10}$ in a mix of water and ethanol in 50/50 proportions in order to obtain a HC$_{10}$ concentration of 30 g/L. Two dipping times were presently considered, a short time of one minute and a prolonged duration of thirty minutes. Samples were then dried at air for fifteen minutes before being introduced in the oven at 80 °C for the night both to dried the poral network efficiently and avoid any seepage during vacuuming samples but also to efficiently eliminate HC$_{10}$ residues inside the porous network and make the samples compatible with vacuum and ultra-high vacuum requirement for SEM-EDS and SAM analyses.

### 2.3. Analyses Methodology

In order to observe the brochantite/cuprite/metal stratification and their corresponding interfaces, the samples were analyzed in cross-section using a multi-technique approach bringing complementary morphological and chemical information from nanometer to micrometer scale. First, optical microscopy and μRS experiments were realized before HC$_{10}$ treatment to bring an overall description of the CPL layout in depth and determine the nature of the CPL. To facilitate analyses interpretation, polished surface was prepared as follows: Epoxy-resin embedding, cutting and surface polishing with SiC papers (180–4000) and diamond spray (1 μm) under deionized water. Then, for the study of the penetration of the HC$_{10}$ treatment in the CPL, SEM-EDS and SAM techniques were used. EDS was employed to assess if Cu(C$_{10}$)$_2$ was present inside the CPL layer and identify areas of interest where SAM, with its high resolution (around 10 nanometer), could evidence the exact penetration depth of the HC$_{10}$ solution. A different preparation mode was required for EDS and SAM elemental mappings, especially not to confuse the carbon arising from the resin with the one coming from the HC$_{10}$ treatment and to minimize topographic artifacts. In this case, the samples were prepared without embedding, using a cross-section polisher tool (JEOL IB-09010CP) according to this protocol: a first mechanical polishing with SiC up to grade 4000 using the JEOL Handy Lap and a finishing stage to obtain a mirror-like surface by Ar$^+$ ion abrasion for 3 h in tangential mode at 4 kV for 3 h (ion current = 65 μA). Thanks to those tailored sample preparations, samples could be analyzed without metallization or charge compensation.

### 2.4. Analytical Techniques

#### 2.4.1. μRaman Spectroscopy (μRS)

μRS measurements were carried out via an Invia Reflex® spectrometer. The incident laser wavelength was of 532 nm and the laser power was filtered down to 0.1 mW. The beam diameter as

well as the probed depth were of about 1 μm. The wavenumber resolution of the spectra was of 2 cm$^{-1}$. The wavenumber calibration of the spectra was obtained from a silicon wafer on the 520.5 cm$^{-1}$ peak. Acquisition and treatment of the spectra were realized with the software Wire 3.4$^{®}$. The reference phases analyzed were commercial cuprite $Cu_2O$ from Sigma Aldrich and brochantite synthetized according to the procedure described by Kratschmer et al. [18]. The synthesis product was pure brochantite, as confirmed by the XRD diffractogram obtained by Apchain [13]. The Raman spectra are presented without smoothing or line fitting and normalized to help comparison along.

### 2.4.2. Scanning Electron Microscopy – Energy Dispersive Spectroscopy (SEM-EDS)

SEM micrographs in secondary electron mode and elemental analyses (EDS) were performed using a JEOL JSM 7001F microscope with a patented "in-lens" Schottky Field Emission Gun (FEG) equipped with an OXFORD Aztec EDS system. The experiments were realized with an incident electron beam of 15 kV accelerating voltage and a working distance of 10 mm between the incident beam and the sample surface. C, O, S Kα et Cu Lα lines were considered to realize the chemical maps. Constructor standard database was used for overall quantification when it was needed.

### 2.4.3. Scanning Auger Microscopy (SAM)

Auger characterizations were performed with a JEOL JAMP 9500F Auger nano-probe also equipped with a patented "in-lens" Schottky Field Emission Gun (FEG) and a hemispherical analyzer (HAS). The ultimate resolutions specified of this equipment are 3 nm (25 kV, 10 pA) for SEM and 8 nm (25 kV, 1 nA) for SAM. Experiments were carried out at 25 kV, 25 nA, tilt 40° leading to 12–15 nm spot size, the maximal analyzed depth being inferior to 4–5 nm. No charge compensation leading to peaks enlargement and energy shifts was used in experiments. In this study, SAM was employed to determine the spatial distribution of the elements inside the CPL with a nanoscale resolution. Local chemical analyses at the interfaces, with a direct access via cross-sections, are presently possible thanks to the inherent beam size of the technique and the quality of the cross-section obtained after ionic polishing, limiting roughness artifacts during SAM measurements. Samples were rapidly transferred to the introduction chamber of the spectrometer just after ion polishing to limit the carbon contamination and oxidation at the surface, detrimental for SAM analyses, which are extremely surface sensitive.

## 3. Results

### 3.1. Nature and Depth Organization of the Corrosion Product Layer

Figure 1a presents the optical microscopy image obtained on the cross-section of a corroded copper sample issued from the roof of the Saint Martin church in Metz and the μRS mapping of the CPL in the selected region. On the optical image, the contrasts clearly enable to separate the CPL (dark contrast) from the copper substrate (light contrast) on which it has grown during air ageing. The CPL is also constituted of two different phases as shown by the two distinct dark contrasts at the metal/CPL interface and at the surface. The typical μRS spectrum evidenced for the inner layer corresponds to cuprite $Cu_2O$ through its main peak at 216 cm$^{-1}$ attributed to the Cu-O bonding (Figure 1c). The characteristic spectrum of the outer layer is attributed to brochantite $Cu_4SO_4(OH)_6$ well characterized by its main peak at 973 cm$^{-1}$ corresponding to the symmetric stretching band S-O (Figure 1b). The typical μRS spectrum of the inner layer corresponds to cuprite $Cu_2O$ through its main peak at 216 cm$^{-1}$ attributed to the Cu-O bonding (Figure 1c). The μRS mapping Figure 1a of cuprite (using the main peak at 216 cm$^{-1}$) and brochantite (using the main peak at 973 cm$^{-1}$) evidences that the entire inner layer (red) is composed of cuprite whereas the thicker outer layer (green) is only made of brochantite. Our observations are in agreement with the literature: these copper corrosion products have already been reported in studies related to the atmospheric corrosion of copper [19–21].

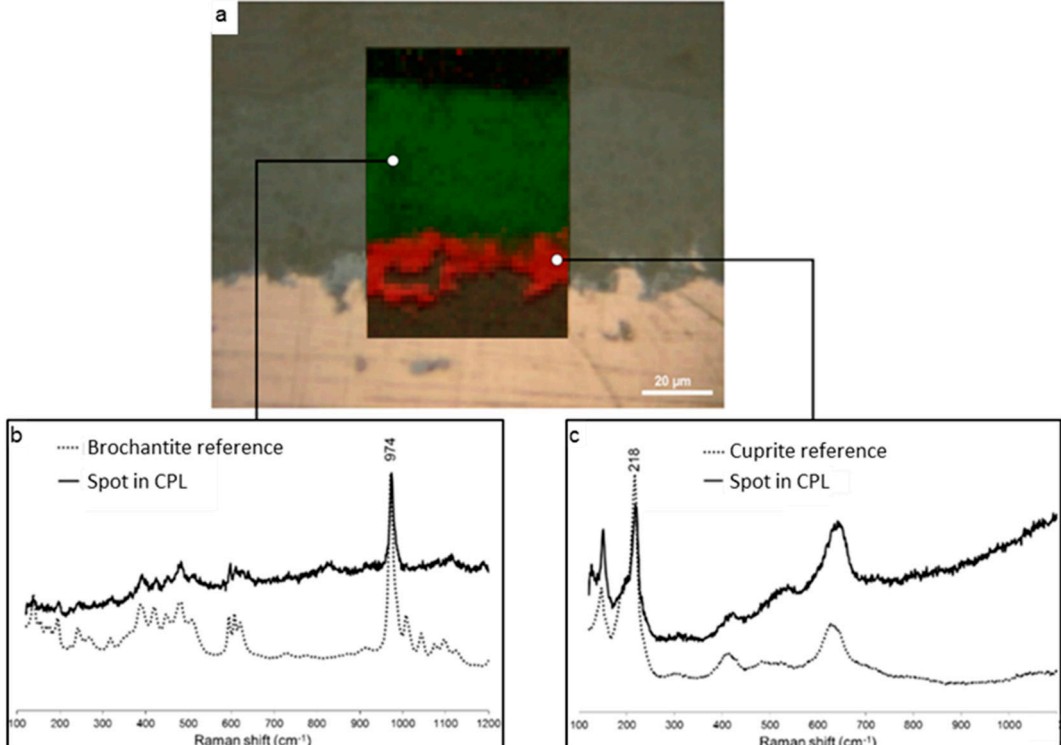

**Figure 1.** Optical microscopy image of the CPL grown on the Cu substrate and µRS mappings of brochantite (green) and cuprite (red) phases (**a**) and their respective corresponding spectra (**b**,**c**).

Figure 2 displays the large scale SEM image of the copper sample cross-section where the different phases of the CPL presented previously are even more visible and well preserved by the ion abrasion, pores, grains and inclusion are revealed. Inner cuprite layer of 5–10 µm thickness at the interface with the metal is rather compact although the brochantite outer layer of 20–50 µm thickness is much more porous.

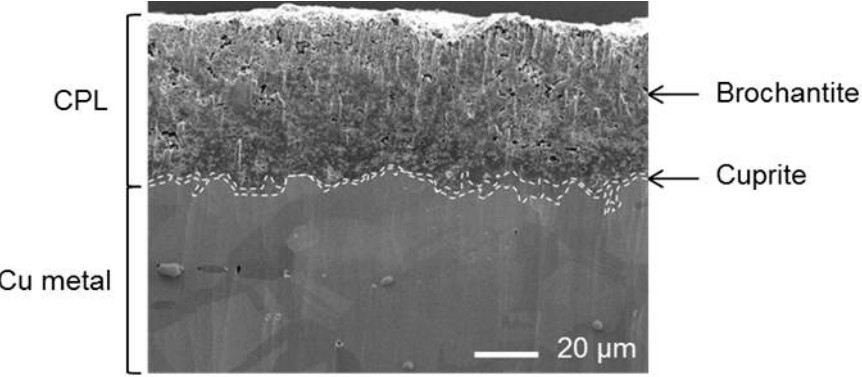

**Figure 2.** SEM image in secondary electron mode.

The SEM-EDS elemental mappings of C, Cu, O and S at the CPL/metal interface region of the sample before $HC_{10}$ immersion are presented on Figure 3. The different phases, metal, cuprite and brochantite, are evidenced on the cartographies and their spatial distribution is distinguishable thanks to the different elements intensities measured and the presence of probe elements. Thus, the presence of brochantite, $Cu_4SO_4(OH)_6$, in the external part of the CPL is supported by the S mapping showing S is only detected in the outside part of the CPL. In accordance with the relative formula of the different layers, such as Cu, $Cu_2O$ and $Cu_4SO_4(OH)_6$, the variable intensities on the Cu mapping are

characteristic of metal (highest intensity), of cuprite (intermediate intensity) and of brochantite (weakest intensity). Conversely, the variable intensities on the O mapping are characteristic of metal (absence of intensity), of cuprite (intermediate intensity) and of brochantite (highest intensity). An inclusion composed of Pb and As is also visible. Even before $HC_{10}$ treatment, the C, which will be the probe element of $HC_{10}$ is detected. Here, this C is inherent to the adventitious superficial contamination deposited on the surface in contact with the atmosphere during the transfer from the CP to the SEM. This carbon contamination is rather homogeneous and mainly mimics the topography, the inclusion is harder and higher and appears in a lighter contrast on contrary to the pores. Note that C sticking slightly differs on the CPL (around 12 at.%) and the metal (around 25 at.%) as already observed by SAM but constant in both areas. This C map presents then the background noise which will be present on all the EDS C maps and thus the detection limit of the $Cu(C_{10})_2$. These results also guarantee that no other carboxylate compounds are detected.

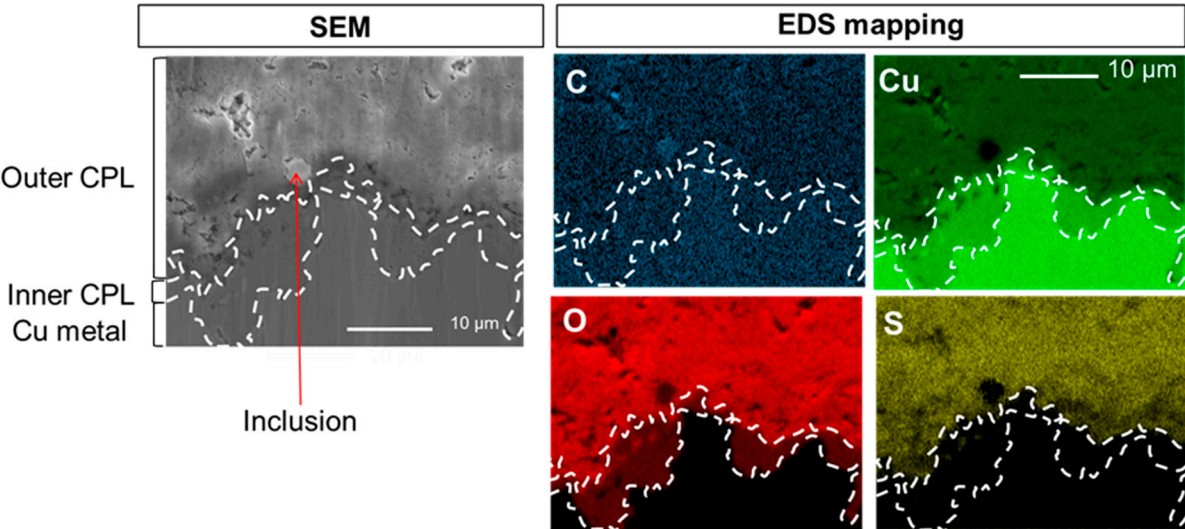

**Figure 3.** SEM image in secondary electron mode and EDS elemental mappings of C, Cu, O and S at the CPL/metal interface.

### 3.2. Penetration of the Carboxylate Treatment in the Corrosion Product Layer

In this study, the penetration of the carboxylate treatment in the corrosion product layer was studied for two immersion times: one minute and thirty minutes. Figure 4 shows pictures of the surface of the samples without immersion and after an immersion of one and thirty minutes in in $HC_{10}$.

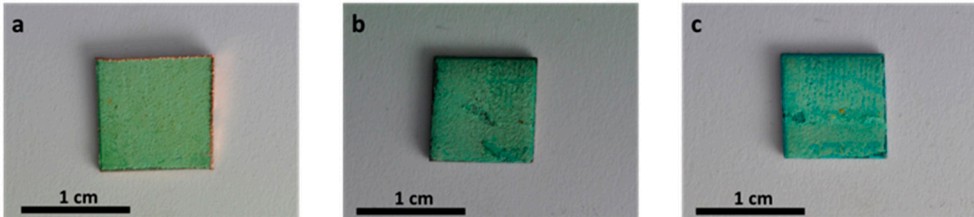

**Figure 4.** Pictures of the samples surface without and with immersion in $HC_{10}$ for different times: No treatment (**a**), 1 minute (**b**) and 30 min (**c**).

The results obtained for the longer immersion time are presented first as the trends are more pronounced.

### 3.2.1. Immersion Time in HC$_{10}$ Solution: Thirty Minutes

First, the SEM-EDS elemental mappings of C, Cu, O and S at the CPL/metal interface region of the sample after 30 min in HC$_{10}$ solution are shown on Figure 5. As for the sample studied previously, the different CPL layers are clearly identified (dotted lines). Focusing on C, the map presents more visible heterogeneities with higher intensities (until almost 50 at.%) than the one measured on the map before treatment. This clearly evidences C incorporation in specific zones of the brochantite layer, porous, but not in the cuprite layer. To clearly identify whether the HC$_{10}$ treatment may start penetrating in the cuprite layer, SAM is used to reach chemical information at the nanometer scale and finely determine the Cu(C$_{10}$)$_2$ formation. Indeed, with EDS, we have succeeded in separating the different phases, delimited by the dotted lines on the maps Figure 5, but with this technique the elemental distribution and especially the one of C for our concern is only accessible at the micrometer scale. A region of interest to realize SAM is identified from EDS mapping (11 µm per 11 µm) and illustrated by the red square on Figure 5.

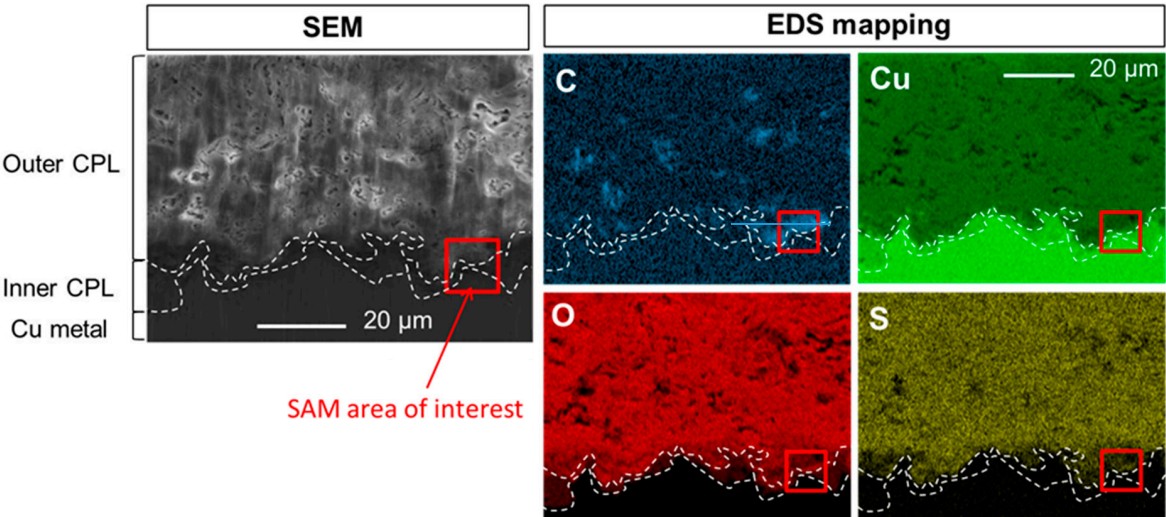

**Figure 5.** Sample immersed in HC$_{10}$ for thirty minutes: SEM image in secondary electron mode and EDS elemental mappings of C, Cu, O and S at the CPL/metal interface.

Figure 6 displays the corresponding magnified SEM image of the region of interest Figure 5, the SAM chemical mapping area of 5.3 µm per 4.9 µm and the overlapping of C and Cu SAM images (C in red and Cu in green) in this area presenting the intensities variation of the C-KLL (263 eV) and Cu-LMM (914 eV) Auger lines. To facilitate the tracking of C within the CPL, the interfaces metal/cuprite and cuprite/brochantite are also indicated by dotted lines on these images. Interpretation of SAM chemical images is simplified when the data acquisition is performed on flat surfaces, enabling to get rid of topography artifacts on the intensities collection (higher intensities in height) and ensure that the intensities variation are only linked to composition modulations. Here, the ion polishing drastically limits such roughness artifacts, and C localization is mainly evidenced in porous areas of the brochantite, not perfectly flat due to this texturation, and the crack observed inside the cuprite on the right side. Auger characterization confirms at the nanometer scale (1pixel ~ 10 nm) what is observed by SEM-EDS at the micrometer scale: carbon is accumulated inside the brochantite pores and the HC$_{10}$ treatment stops at the cuprite/brochantite interface.

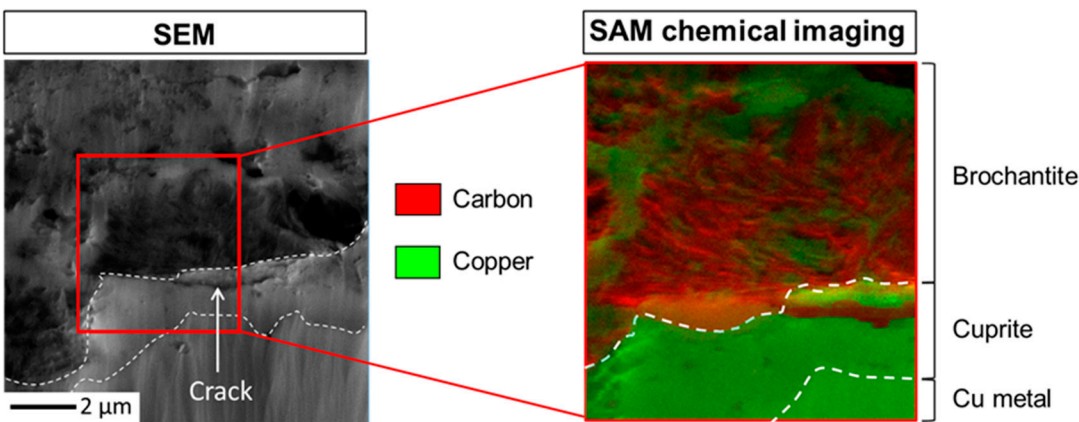

**Figure 6.** Sample immersed in $HC_{10}$ for thirty minutes: SEM image and SAM analysis area (5.3 μm per 4.9 μm) (**a**), overlapping of C and Cu chemical images (**b**).

### 3.2.2. Immersion Time in $HC_{10}$ Solution: One Minute

Shorter treatment duration has also been studied to investigate the penetration kinetic of $HC_{10}$ solution, and so, optimize the procedure. The SEM-EDS elemental mappings of C, Cu, O and S in the CPL of the sample after its immersion in $HC_{10}$ for one minute are displayed on Figure 7. The CPL layers are still clearly identified (dotted lines) and the C seems to be only detected in the brochantite layer. However, it seems less abundant than in the case of the immersed sample for thirty minutes. The red square corresponds to the selected region of interest of 29 μm per 29 μm for complementary SAM fine characterization.

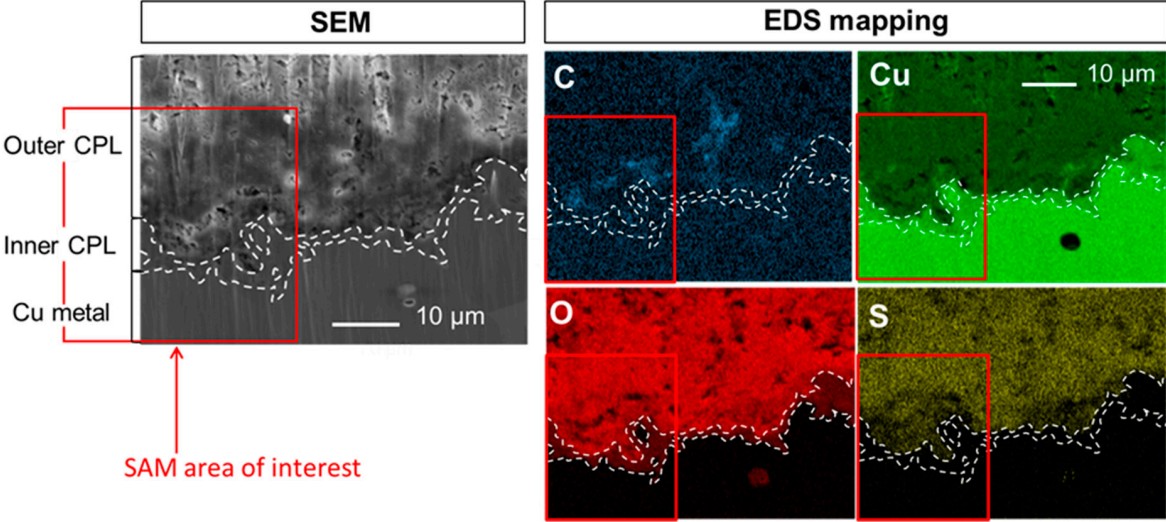

**Figure 7.** Sample immersed in $HC_{10}$ for one minute: SEM-EDS elemental mappings of C, Cu, O and S at the CPL/metal interface.

Figure 8 shows the corresponding magnified SE image of the square zone Figure 7, the area for SAM cartography analyses (13.3 μm per 2.2 μm) and the overlapping of C and Cu chemical images (Cu in green and C in red) in this area. The interfaces metal/cuprite and cuprite/brochantite are also indicated on these images (dotted lines). As for the immersion time of thirty minutes, C is only seen in the brochantite layer, principally in the porous parts and stops once again at the brochantite/cuprite interface. This result indicates that the $HC_{10}$ treatment infiltration is fast, but, after one minute, only partial since conversely to the 30 min dipping C is more locally detected inside the brochantite and at the brochantite/cuprite interface.

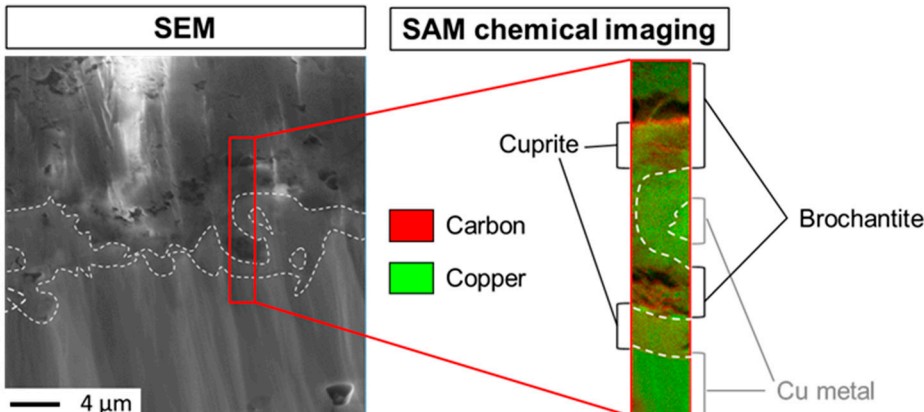

**Figure 8.** Sample immersed in $HC_{10}$ for one minute: SEM image and SAM analysis area (13.3 μm per 2.2 μm) (**a**), overlapping of C and Cu chemical images (**b**).

## 4. Discussion

The analytical methodology employed in this paper highlights the complementarity of the analysis techniques chosen, enabling to bring an accurate morphological and chemical characterization of the penetration depth of $HC_{10}$ treatment in the CPL on copper metal. All the techniques selected present spatial resolutions adapted to the direct analysis on cross-sections, preventing using depth profiling techniques. This brings the advantage to take account for the complex brochantite/cuprite and cuprite/metal interfaces inherent to the irregular CPL growth at the Cu surface (different thicknesses, pores distribution, initial metal surface irregularities, corrosion process … ). Thanks to the combine use of μRS, SEM-EDS and SAM, the nature of the CPL and its depth organization can be identified and the $HC_{10}$ treatment localization can be determined from the micrometer to the nanometer scale. The benefit of the specific cross-section surface preparation by ionic polishing is evidenced since it perfectly reveals the microstructure and interfaces without requiring the use of epoxy resin, that would get into the porous layer of brochantite during the polishing step and hinder the interpretation of the chemical mappings of carbon in CPL whose origin could either be $HC_{10}$ or resin. In addition, topographical artifacts can be satisfactorily limited. As mentioned in the introduction, such methodological approach, combining complementary techniques for a complete characterization, has already been employed to study the corrosion mechanism of many Cu based artefacts bringing key information for the safeguard [18,19,21]. This paper is another illustration of its real added-value also for the development of new preservation strategies based on the use of innovative corrosion inhibitors presenting lower toxicity.

The results evidence that the corrosion product layer of the copper samples issued from the roof of Saint Martin church in Metz is composed of an inner layer consisting of cuprite covered by a thicker and a more porous outer layer of brochantite, as previously reported in [13]. This two-layer structure is common for copper corrosion and is found in various studies dealing with the atmospheric corrosion of this metal [18,22–24]. The presence of an outer sulfur layer of brochantite ($Cu_4SO_4(OH)_6$) is characteristic of an exhibition in the outdoor environment. Indeed, the formation of the CPL occurs in two stages: the corrosion of copper metal which leads to the precipitation of cuprite ($Cu_2O$) and, when the layer of cuprite is stabilized in thickness [22], the formation of a layer of Cu(II) depending on species present in the atmosphere and in the aqueous film on the surface of the object [18,19,25–28]. In standard atmospheric conditions Cu(II) ions react with $SO_2$ and $SO_4$ ions present in the atmosphere [29] and a brochantite layer is formed on top of the cuprite one.

Concerning the penetration of the $HC_{10}$ treatment in the CPL, for the two immersion times studied, one and thirty minutes, SEM-EDS and SAM demonstrate that the treatment is concentrated in the pores of the brochantite layer and is blocked at the brochantite/cuprite interface, without clear evidence of diffusion in the cuprite layer. The treatment duration of the sample has an effect on the $HC_{10}$ penetration in the CPL as for the immersion time of one minute, the $HC_{10}$ distribution is more

local than for the immersion time of thirty minutes where the treatment is distributed in the whole layer of brochantite. The fact that the $HC_{10}$ treatment stops at the cuprite/brochantite interface raises the question of the inhibition effect of the $HC_{10}$ but also of the permeability of the cuprite layer.

Many studies suggest the existence of protective properties of the inner layer of cuprite [22,30–32]. This "barrier" effect of the cuprite layer has been put forward previously by the stabilization of the thickness of the cuprite layer over time [22]. We have also demonstrated elsewhere [13], on the base of observations of the copper corrosion by different techniques (optical microscopy, SEM), the lack of microscopic porosity of the cuprite layer compared for instance to the much more porous brochantite outer layer. In addition, for copper samples treated by immersion in $HC_{10}$ solutions, μRS has evidenced the absence of the main peak characteristic of $Cu(C_{10})_2$ in cuprite layers whereas it was visible in brochantite layers. Furthermore, the $HC_{10}$ reacts with corrosion products to form $Cu(C_{10})_2$, suggesting that the $HC_{10}$ treatment penetration must be related not only to the porosity properties of cuprite but also to the low reactivity properties of cuprite with $HC_{10}$ which probably don't dissolve so easily even in local acidic conditions. From our expertise and the state of the art in the bibliography, the cuprite layer nature influences the treatment efficiency by hindering the penetration of the $HC_{10}$ treatment. Present results demonstrate that the treatment does not reach the cuprite/metal interface where the anodic reaction governing the corrosion process occurs but the cuprite layer itself, due to its low porosity and chemical reactivity, prevent corrosion phenomenon by forming a protective barrier on the metal. However, as cuprite dissolution may nonetheless occur in some cases, the use of an additional chemical treatment ensures an efficient protection of the object whatever the circumstances.

Finally, these results bring a deeper knowledge on the interaction mechanism between CPL and $HC_{10}$ corrosion inhibitor, demonstrating that $HC_{10}$ solution penetrates rapidly inside the brochantite layer (less than 1 min to reach the brochantite/cuprite interface), by means of the numerous pores, leading to the progressive formation of $Cu(C_{10})_2$ inside the pores. After 30 min, the main part of the pores seems to contain organic precipitate and $Cu(C_{10})_2$ is also clearly detected at the brochantite/cuprite interface where the $HC_{10}$ is stopped. Indeed, the inner cuprite layer appears free from $Cu(C_{10})_2$. This emphasizes the natural barrier effect of the inner cuprite layer, compact and stable, on top of the metal. This also opens interesting perspectives for the protection of copper objects, the major issue concerning the interaction of inhibitors with brochantite to stabilize this layer and prevent infiltration. Different wet organic treatments can be envisaged, to selectively interact with brochantite and form a "second barrier layer" on top of the native cuprite one, while conserving the aesthetic aspect of this outer layer. Due to the ease of use, fast reaction and non-toxicity of these treatments, unlike waxes, varnishes and benzotriazole-type corrosion inhibitors, we show that this type of carboxylate treatment could be used on Cu heritage objects a larger scale. New developments are ongoing as well as ageing studies.

## 5. Conclusions

An approach based on the use of complementary analytical methods from micrometer to nanometer scale (μRS, SEM-EDS, SAM) is developed in this study to describe the properties of the CPL and the penetration of the carboxylate ($HC_{10}$) treatment in the CPL of copper samples issued from the roof of Saint Martin church in Metz exposed to atmospheric corrosion. The CPL of the samples consists of a thin layer (1 to 5 μm thick) of cuprite $Cu_2O$ on the surface of the metal, followed by a second thicker layer (20 to 50 μm), mainly composed of $Cu_4SO_4(OH)_6$ (brochantite). The short-term immersion tests (one minute) with $HC_{10}$ shows that the protection treatment penetrates to the cuprite/brochantite interface but is only detected locally, whereas for a longer immersion time of thirty minutes it is present in the whole brochantite layer, especially in porous regions, and also up to the cuprite/brochantite interface where it appears blocked. In present conditions, the cuprite layer is by consequence not impacted by the carboxylate treatment and these results evidence at the same time the protective properties of this inner layer for the metallic Cu beyond. This indicates also that $HC_{10}$ selectively interacts with brochantite, where $Cu(C_{10})_2$ precipitates inside the pores and interstices limiting and preventing further liquid infiltration and corrosion phenomenon. The upper part of the CPL acts as

an additional barrier layer, reinforcing the protective action of the inner cuprite layer. These results enable to better understand the $HC_{10}$/CPL interaction and evidence that the use of organic inhibitor is an interesting route to consider for conservation and preservation issues, less toxic and with possible large scale application.

**Author Contributions:** Conceptualization, M.B., F.M.-B., D.N., E.A. and M.L.; methodology, M.B., F.M.-B., M.L. and E.A.; software, M.B., F.M.-B., M.L., E.A. and D.N.; validation, M.B., F.M.-B., M.L., E.A. and D.N.; formal analysis, F.M.-B., E.A., M.B., M.L.; investigation, M.B., F.M.-B., M.L., E.A. and D.N.; resources, D.N., P.D., A.E., M.B., and E.A.; data curation, M.L., E.A., M.B. and F.M.-B.; writing—original draft preparation, F.M.-B.; writing—review and editing, F.M.-B., M.B., M.L, D.N. and E.A.; visualization, M.B., F.M.-B., M.L., E.A., D.N., A.E and P.D.; supervision, F.M.-B., M.B., M.L., E.A. and D.N.; project administration, D.N., P.D. and E.A.; funding acquisition, D.N., P.D. and E.A.

**Acknowledgments:** The authors wish to thank Annick Texier and Aurélia Azéma of the Laboratoire de Recherche sur les Monuments Historiques (LRMH) for providing us the samples from the roof of Saint Martin church in Metz. This work was carried out thanks to a financial support of Fondation des Sciences du Patrimoine.

**Conflicts of Interest:** The authors declare no conflict of interest.

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
