# Peer review of "Multiscale Study of Interactions Between Corrosion Products Layer Formed on Heritage Cu Objects and Organic Protection Treatments"

_heritage, doi:10.3390/heritage2030162_

Round 1

Reviewer 1 Report

A few comments are included in the PDF file

Author Response

Dear Editor, dear Reviewers,

            We would like to thank you for the high interest and your accurate review of our work. Your constructive comments have helped us to improve the paper.

Below you can find our point-by-point responses under each comment made by the respective reviewer along with the corresponding changes as indicated.

            With our best wishes and on behalf of all the co-authors,

Florence Mercier

Corrections

---------------------------------------------------

The manuscript has been carefully re-read with a special attention for verb tenses. Modifications have been made all along the text to fulfill the reviewers recommendations to emphasize in particular the novelty of the work, how it helps to get a better understanding on the HC10 protection mechanism, and the consequences of the results presented for the cultural heritage safeguard.

A paragraph has been added at the end of the introduction to better introduce the benefit and challenges of the multi-technique characterization and make a link with the discussion part; new references have been added in agreement with reviewer 3 comment.

A modification in the structuration of the article has also been done to improve clarity in the “Materials and method” section divided now in 4 paragraphs (samples, carboxylate protective treatment, analyses methodology and analytical techniques) and in the results part where all the information regarding the sample before treatment have been gathered in the same paragraph “Nature and depth organization of the corrosion product layer”. The Figure 3 corresponds now to Figure 2 and EDS maps of the sample before treatment has been added as recommended by reviewers 2 and 3 (corresponding to Figure 3) with a descriptive paragraph. The Figures 1 and initially 2 (now Figure 4) have been modified and uploaded with a better resolution. The Figures numbering has been changed accordingly in the whole paper.

All the modifications are underlined in yellow in the manuscript.

Reviewer 1

The few comments included in the PDF file have been corrected in the manuscript.

Reviewer 2

Specific comments:

Page 2, Line 33. Please correct: To bring new insights on these this problematic. Corrected

Page 3, Line 1. Please correct: Analyses are were realized. As mentioned above, this grammar error is frequent throughout the manuscript and should be corrected. Other examples in Page 3, Line 7: The samples are were prepared; Line 10: They are were cleaned; Line 12: The protective treatment is was realized; Line 13: The solution is was prepared; Line 14: Two dipping times are presently were considered; Line 16: Samples are were then dried; Line 19: the samples are were analyzed. Corrected

Page 3, Line 9: The authors detail the composition of the copper metal as “containing micrometric inclusions mostly composed of lead, arsenic, but also antimony, nickel and bismuth”. The analytical technique that enabled this characterization should be mentioned.

The analytical technique that enabled this characterization is SEM-EDS and this information was added in the manuscript.

Page 3, Line 16: The authors refer that the copper samples were introduced “in the oven for the night”. What was the temperature used and what is its impact?

Samples were then dried at air for fifteen minutes before being introduced in the oven at 80°C for the night both to dried the poral network efficiently and avoid any seepage during vacuuming samples but also to efficiently eliminate HC10 residues inside the porous network and make the samples compatible with vacuum and ultra-high vacuum requirement for SEM-EDS and SAM analyses.

Page 3, Line 19: Please delete “preliminary prepared as describe later on”. This paragraph dedicated to the preparation of the samples should be better described. It is also suggested to replace ionic abrasion for ion abrasion throughout the manuscript.

Corrected and this paragraph has been modified to improve clarity about the two kinds of sample preparation realized according to the analyses techniques requirements.

Page 4, Line 9: What was the synthesis method used to produce brochantite? A reference should be added to support this information.

The synthesis of brochantite was made according to the procedure described by Kratschmer et al. (A. Kratschmer, I. Wallinder, C. Leygraf, The evolution of out copper patina, Corros. Sci. 44 (2002) 425–450). This sentence and the reference associated to the brochantite synthesis were added in the manuscript.

And what was the purity of the sample produced? Was it analyzed by another technique such as X-ray Diffraction?

The synthesis product was pure brochantite, as confirmed by the XRD diffractogram obtained by Apchain (Apchain, E. Apport des traitements carboxylates à la protection des alliages cuivreux, Thèse de l’Université de Cergy-Pontoise 2018). This sentence was added in the manuscript.

Page 4, Line 25: Please delete “preliminary prepared”.

Corrected

Page 4, Line 27: Please substitute: “Samples are rapidly transferred in the introduction chamber of the spectrometer just after ionic polishing to limit the carbon contamination and oxidation at the surface, detrimental for AES analyses extremely surface sensitive.” for “Samples were rapidly transferred to the introduction chamber of the spectrometer just after ion polishing to limit the carbon contamination and oxidation at the surface, detrimental for AES analyses, which are extremely surface sensitive.”

Corrected

Page 5, Line 16: Please add reference: Leygraf, C.; Wallinder, I.O.; Tidblad, J.; Graedel, T. Atmospheric Corrosion, 2nd ed.; John Wiley & Sons: Hoboken, NJ, USA, 2016. It is a key reference when studying the atmospheric corrosion of copper.

The reference was added in the manuscript

Page 6, Figure 1: The figure should be re-organized and have a higher resolution. The image in the middle is not necessary; the analyzed points can be drawn in figure a); the numbers of the Raman spectrum, including the wavenumbers, are unreadable, and their reading is essential in a scientific publication.

The figure 1 was replaced in the manuscript.

Page 7, Figure 2: The effect of the treatment is not clear at all. The figure should have a higher resolution and have less blank space; should be more focus on the copper samples.

The figure 2 was replaced in the manuscript.

Page 7, Figure 3: In order to complement Figure 3, it is recommended to add the elemental mappings of C, Cu, O and S of the cross-section before the HC10 treatment, in the Appendix section, so the readers may compare with the others where the HC10 was applied.

The EDS maps obtained before HC10 immersion, also asked by reviewer 3, have been added in the body of the text (Figure 3).

Page 8, Line 20: Please correct “micrometrer scale” for micrometer scale.

Corrected

Page 11, Line 31: The detection of the Cu(C10)2 by Raman microscopy should be presented and its role in the preservation of the patina should be re-addressed, linking the discussion to the introduction.

In the thesis of Apchain (Apchain, E. Apport des traitements carboxylates à la protection des alliages cuivreux, Thèse de l’Université de Cergy-Pontoise 2018), the presence of Cu(C10)2 in the CPL was observed by µRaman from the presence of the peak at 288 cm-1. These µRaman data are currently the subject of another article in preparation.

The authors should state very clearly how this work advanced the current knowledge on the CPL/HC10 layer interaction mechanism and the effectiveness of the HC10 treatment.

A paragraph has been added at the end of the discussion part.

It is recommended to use consistent terms and acronyms throughout the manuscript. For example: µRS for Raman analysis, SE for Scanning Electron micrographs (remove SEI), EDS for elemental analysis complementing SE micrographs, SAM for Scanning Auger micrographs (replace AES area of interest for SAM area of interest). Consistency brings clarity to the manuscript.”

Corrected. Consistent terms and acronyms were used throughout the manuscript.

Reviewer 3

1) More references should be provided in the introduction. In particular, there are several papers in the literature that report a multi-analytical approach based on the use of complementary techniques (i.e. SEM-EDS, ATR-FTIR, XPS, XRD, DTA-TG, ICP-MS, Tof-SIMS and optical microscopy) for the study of the corrosion products of copper-based materials and metallic artefacts:

-Applied Surface Science 470 (2019) 74–83 

-Applied Surface Science 446 (2018) 168–176

-Applied Surface Science 470 (2019) 695–706

The three references concerning the application of a multi-analytical approach based on the use of complementary techniques for the study of the corrosion products of copper-based materials and metallic artefacts were added in the manuscript. A linked sentence was added in the manuscript. We thank the reviewer for those very interesting and recent papers.

2) The quality of Figure 1 and of Figure 2 should be improved

Figures 1 and 2 were replaced in the manuscript.

3) The SEM-EDS elemental mappings of C, Cu, O and S at the CPL/metal interface region of the sample before immersion in HC10 have be shown as reference. It is important to prove that the presence of C do not derives from carbonate-based product but only from HC10

The EDS maps obtained before HC10 immersion, also asked by reviewer 2, have been added in the body of the text (Figure 3).

The presence of C is detected at a similar level (roughly constant intensity) on the CPL and the metal, the sticking coefficient being different on CPL (12 at.%) and metal (25 at.%). This C corresponds to the adventitious C contamination present at the sample surface, and can be considered as our detection limit as well as our background noise.  The influence of the topography on the intensity measurement is rather improbable or limited since the surface was polished with the CP. Consequently, the higher contrasts observed on the C map after treatment are definitely attributed to the Cu(C10)2 formation.  Indeed, HC10 has been removed after rinsing and heating in the furnace, before sample polishing, leaving only the precipitates formed. No other carbonate based compounds have been found.

Reviewer 2 Report

Dear editor and authors, as a reviewer, I recommend that the manuscript entitled “Multiscale study of interactions between corrosion products layer formed on heritage Cu objects and organic protection treatments” may be accepted for publication in Heritage after minor revisions.

Through an analytical approach based on the complementary techniques Raman Microscopy, Secondary Electron Microscopy/Electron Dispersive Spectroscopy and Scanning Auger Microscopy, the authors studied the impact of a new corrosion inhibitor based on a carboxylate treatment (HC10) on the corrosion products layer (CPL) of copper samples issued from the roof of the Saint Martin church in Metz. They investigated the optimization of the HC10 protection treatment on a CPL composed of a thick brochantite layer (20 to 50 μm) on top of a thinner cuprite layer (1 to 5 μm), localized at the interface with the copper metal. The characterization of the penetration depth allowed to demonstrate that long immersion times of the HC10 solution in the CPL result in a better penetration. Nonetheless, the cuprite layer hindered the penetration of the HC10 treatment, which also shows its protective properties. These results are of great interest to those interested in the protection of copper cultural heritage objects and to the scientific community studying the degradation phenomena of copper objects caused by atmospheric corrosion.

In general, the manuscript is clear, well organized and discussed. However, a revision by an English speaker is recommended as there is frequent incorrect use of verb tenses, which disturbs the reading of the manuscript. It would also benefit if the abstract described better the novelty of the work. The implication of the results towards the safeguard of copper cultural heritage objects should be more emphasised in the discussion and conclusions sections. Furthermore, the large scale application should also be addressed as it is mentioned in the abstract.

Overall, the figures are well-presented, however, Figures 1 and 2 should be revised as described below. All figure captions should be simplified; it is suggested to remove the details on the preparation of the cross-sections and of the analyses; this should be well detailed in the methods section. A figure caption should be clear and succinct.

It is recommended to use consistent terms and acronyms throughout the manuscript. For example: μRS for Raman analysis, SE for Scanning Electron micrographs (remove SEI), EDS for elemental analysis complementing SE micrographs, SAM for Scanning Auger micrographs (replace AES area of interest for SAM area of interest). Consistency brings clarity to the manuscript.

SPECIFIC COMMENTS

Page 2, Line 33. Please correct: To bring new insights on these this problematic.

Page 3, Line 1. Please correct: Analyses are were realized. As mentioned above, this grammar error is frequent throughout the manuscript and should be corrected. Other examples in Page 3, Line 7: The samples are were prepared; Line 10: They are were cleaned; Line 12: The protective treatment is was realized; Line 13: The solution is was prepared; Line 14: Two dipping times are presently were considered; Line 16: Samples are were then dried; Line 19: the samples are were analyzed.

Page 3, Line 9: The authors detail the composition of the copper metal as “containing micrometric inclusions mostly composed of lead, arsenic, but also antimony, nickel and bismuth”. The analytical technique that enabled this characterization should be mentioned.

Page 3, Line 16: The authors refer that the copper samples were introduced “in the oven for the night”. What was the temperature used and what is its impact?

Page 3, Line 19: Please delete “preliminary prepared as describe later on”. This paragraph dedicated to the preparation of the samples should be better described. It is also suggested to replace ionic abrasion for ion abrasion throughout the manuscript.

Page 4, Line 9: What was the synthesis method used to produce brochantite? A reference should be added to support this information. And what was the purity of the sample produced? Was it analyzed by another technique such as X-ray Diffraction?

Page 4, Line 25: Please delete “preliminary prepared”.

Page 4, Line 27: Please substitute: “Samples are rapidly transferred in the introduction chamber of the spectrometer just after ionic polishing to limit the carbon contamination and oxidation at the surface, detrimental for AES analyses extremely surface sensitive.” for “Samples were rapidly transferred to the introduction chamber of the spectrometer just after ion polishing to limit the carbon contamination and oxidation at the surface, detrimental for AES analyses, which are extremely surface sensitive.”

Page 5, Line 16: Please add reference: Leygraf, C.; Wallinder, I.O.; Tidblad, J.; Graedel, T. Atmospheric Corrosion, 2nd ed.; John Wiley & Sons: Hoboken, NJ, USA, 2016. It is a key reference when studying the atmospheric corrosion of copper.

Page 6, Figure 1: The figure should be re-organized and have a higher resolution. The image in the middle is not necessary; the analyzed points can be drawn in figure a); the numbers of the Raman spectrum, including the wavenumbers, are unreadable, and their reading is essential in a scientific publication.

Page 7, Figure 2: The effect of the treatment is not clear at all. The figure should have a higher resolution and have less blank space; should be more focus on the copper samples.

Page 7, Figure 3: In order to complement Figure 3, it is recommended to add the elemental mappings of C, Cu, O and S of the cross-section before the HC10 treatment, in the Appendix section, so the readers may compare with the others where the HC10 was applied.

Page 8, Line 20: Please correct “micrometrer scale” for micrometer scale.

Page 11, Line 31: The detection of the Cu(C10)2 by Raman microscopy should be presented and its role in the preservation of the patina should be re-addressed, linking the discussion to the introduction.

The authors should state very clearly how this work advanced the current knowledge on the CPL/HC10 layer interaction mechanism and the effectiveness of the HC10 treatment.

Author Response

(The authors gave the same response as above.)

Reviewer 3 Report

1) More refernces should be provided in the introduction. In particular, there are several papers in the literature that report a multi-analytical approach based on the use of complementary techniques (i.e. SEM-EDS, ATR-FTIR, XPS, XRD, DTA-TG, ICP-MS, Tof-SIMS and optical microscopy) for the study of the corrosion products of copper-based materials and metallic artefacts:

-Applied Surface Science 470 (2019) 74–83 

-Applied Surface Science 446 (2018) 168–176

-Applied Surface Science 470 (2019) 695–706

2) The quality of Figure 1 and of Figure 2 should be improved.

3) The SEM-EDS elemental mappings of C, Cu, O and S at the CPL/metal interface region of the sample before immersion in HC10 have to be shown as reference. It is important to prove that the presence of C do not derives from carbonate-based product but only from HC10.

Author Response

(The authors gave the same response as above.)
